# Significance of Cardiac Magnetic Resonance Feature Tracking of the Right Ventricle in Predicting Subclinical Dysfunction in Patients with Thalassemia Major

**DOI:** 10.3390/diagnostics12081920

**Published:** 2022-08-09

**Authors:** Karuna M. Das, Usama M. A. Baskaki, Anisha Pulinchani, Huthaifa M. Ali, Taleb M. Almanssori, Klaus Van Gorkom, Amrita Das, Hany Dewedar, Sanjiv Sharma

**Affiliations:** 1College of Medicine & Health Sciences, United Arab Emirates University, Al Ain P.O. Box 17666, United Arab Emirates; 2Rashid Hospital, Dubai P.O. Box 4545, United Arab Emirates; 3Department of Data Science, PSPH, Manipal Academy of Higher Education, Manipal 576104, India; 4Brighton College, Al Ain P.O. Box 17666, United Arab Emirates; 5Thalassemia Center, Dubai P.O. Box 9115, United Arab Emirates; 6AIIMS, New Delhi 110029, India

**Keywords:** thalassemia major, CMR-FT, RVGLS, MIO

## Abstract

In patients with thalassemia major (TM), cardiac magnetic resonance feature-tracking (CMR-FT) has been shown to be an effective method for diagnosing subclinical left ventricular (LV) dysfunction. This study aimed to determine whether CMR-FT could detect abnormal RV dysfunction in patients with a normal right ventricular ejection fraction (RVEF). We performed a retrospective analysis of TM patients admitted to Dubai’s Rashid Hospital between July 2019 and March 2021. The inclusion criteria were TM patients with SSFP cine with T2* (T2*-weighted imaging), while exclusion criteria included any other cardiovascular disease. When there was no myocardial iron overload (MIO) (T2* ≥ 20 ms) and when there was significant MIO (T2* < 20 ms), the CMR-FT was used to correlate with EF. Among the 89 participants, there were 46 men (51.7%) and 43 women (48.3%), with a mean age of 26.14 ± 7.4 years (range from 10 to 48 years). Forty-six patients (51.69%) did not have MIO, while 43 individuals did (48.31%). Thirty-nine patients (32.6%) were diagnosed with severe MIO, while seventeen (19.1%) were diagnosed with mild to moderate MIO. A significant correlation existed between RVEF and T2* values (r = 0.274, *p* = 0.014) and between left ventricular ejection fraction (LVEF) and T2* values (r = 0.256, *p* = 0.022). Using a multiple logistic regression model with predictors such as right ventricular longitudinal strain (RVGLS), LV ejection fraction (LV EF), and hemoglobin, abnormal myocardial iron overload can be predicted. This model demonstrates an AUC of 78.3%, a sensitivity of 72%, and a specificity of 76%. In the group with preserved RVEF > 53%, the left ventricular radial strain (LVGRS) (*p* = 0.001), right ventricular radial strain (RVGRS) (*p* = 0.000), and right ventricular basal circumferential strain (RVGCS-basal) (*p* = 0.000) CMR-FT strain values are significantly lower than those of the control group (*p* > 0.05). There was no significant correlation between the LVGLS and T2*. RVGLS was ranked among the most accurate predictors of abnormal myocardial iron overload. The LVGRS, RVGRS, and RVGCS-basal CMR-FT strain values were the best predictors of subclinical RV dysfunction in the group with preserved RVEF. The most accurate way to diagnose MIO is still T2*, but FT-strain can help us figure out how MIO affects the myocardium from a pathophysiological point of view.

## 1. Introduction

Thalassemia results in chronic hemolytic anemia, the most prevalent monogenetic disease in the world, and affected patients frequently need blood transfusions [1]. MIO is a leading cause of heart failure (HF) [2]. Until recently, the left ventricle (LV) received considerable attention in studies of thalassemia major (TM)-associated cardiac dysfunction [3,4]. Recent studies have touted cardiac magnetic resonance feature tracking (CMR-FT) as a game-changing, high-resolution technique for measuring left and right ventricular myocardial deformation [5,6]. Recent CMR research revealed a significant correlation between CMR-FT and T2*-weighted imaging (T2*) and their ability to detect contractile abnormalities in thalassemia patients with normal LVEF [7]. When comparing MIO patients with preserved EF to those without, the left ventricle strain was significantly lower in those with preserved EF, but not much is known about how myocardial iron overload cardiomyopathy affects the right ventricle (RV) in people with thalassemia.

Due to repeated blood transfusions, myocardial deformations change, resulting in variations in strain values. FT-CMR is a highly accurate method for detecting myocardial dysfunction and myocardial fiber deformation, especially in the earliest stages of cardiac disease [8,9]. By evaluating strain values, FT-CMR is a highly accurate technique for detecting myocardial dysfunction and myocardial fiber deformation, especially in the early stages of cardiac disease [10]. Myocardial strain measurements using the CMR feature tracking technique (CMR-FT) have proven to be highly repeatable and reliable in the literature, despite minor differences between software vendors [11]. CMR-FT requires less post-processing time than previous strain analysis techniques, may be less dependent on the operator, and is compatible with standard cine CMR images [11]. Other methods of measuring RV deformation are inferior to FT-CMR because they can not effectively track the right ventricular walls [12].

Recent research indicates that myocardial iron deposition is significantly linked to RV dysfunction, corroborating the decline in LV function associated with increased cardiac iron overload [13,14]. A progressive and significant decrease in RVEF was observed when myocardial T2* was <20 ms. Due to the anatomy and muscle patterns of the right ventricular complex, cardiovascular magnetic resonance (CMR) has been deemed superior to echocardiography for determining the precise RVEF function and chamber size [15,16,17]. To our knowledge, CMR-FT has not been utilized to assess right ventricular dysfunction in TM. Our goal was to see if CMR-FT could be used to tell if the right ventricle was not working right and to measure global and localized right ventricular myocardial deformation in people with TM who had a good RVEF.

## 2. Methods

Between July 2019 and March 2021, we conducted a retrospective analysis of TM patients referred by the Thalassemia clinic for cardiac MR examination to the Rashid Hospital in Dubai. The inclusion criteria included a confirmed diagnosis of TM based on previous measurements [1,18]. Excluded were patients with a history of cardiovascular disease (valvular, congenital, or arrhythmia) and those with cardiovascular risk factors such as diabetes, hypertension, smoking, cardiotoxic medication, or hypothyroidism. Clinical, demographic, and CMR data were examined in the medical records. The Institutional Review Board of the Dubai Health Authority approved the study and waived the need for informed consent (DSREC-02/2020_11, 20 April 2020).

In an anemia-induced high-output state, the ejection fraction of patients with hemoglobinopathies and other hematologic disorders is typically normal or elevated. Consequently, a typical LVEF and RVEF range was determined by averaging the values obtained from our TM cohort in the absence of evidence of myocardial iron overload. The 95% Confidence Interval for the mean LV EF was (56.86, 60.04), and the 95% Confidence Interval for the mean RV EF was (49.77, 53.24). Based on these upper limits of the means of LVEF and RVEF, as it says in the manuscript, we put the patients into two groups: normal and abnormal.

## 3. Control Selection

We conducted a retrospective screening of 24 individuals who had previously been evaluated prospectively at the same facility and were found to be in excellent health. Exclusion criteria included known cardiovascular risk factors, any pre-existing diseases or medications, an impaired left ventricular ejection fraction (LVEF) of less than 55%, or abnormal findings on a 12-lead electrocardiogram or magnetic resonance imaging scan (MRI). Insufficient CMR data for feature tracking analysis led to the exclusion. An ethics committee authorized all of the studies.

## 4. Cardiac MR Image Acquisition

Magnetom Aera was utilized to conduct a standard cardiac MRI (Siemens Medical Systems, Erlangen, Germany). Short-axis, horizontal long-axis, three-chamber view, and vertical long-axis cine series were acquired using a steady-state free precession sequence with typical TR/TE values of 2.2/1.1 ms at 1.5 T (flip angle, 60°). The short-axis cine images consisted of 10–12 sections, each 8 mm thick and separated by a 2 mm gap, spanning from 1 cm above the mitral valve plane to the apex. T2* was measured using a single breath-hold multi-echo gradient-echo sequence. As previously described, the scanning for myocardial T2* measurement was synchronized to the cardiac cycle using standard ECG gating [19]. A 10 mm thick short-axis mid-ventricular slice at the level of the papillary muscles was also performed.

All CMR data were retrieved and analyzed on a computer (Cvi42; Circle Cardiovascular Imaging, Inc., Calgary, AB, Canada). This method is based on an incompressible volume-based method that has been previously validated for accurate biventricular anatomical monitoring. This method is used for biventricular anatomical monitoring [20]. An experienced radiologist manually determined the endocardial and epicardial borders in serial short-axis slices taken at the end-diastolic and end-systolic phases. Using serial short-axis slices, the right ventricular geometry and function parameters were determined, including right ventricular end-diastolic volume, right ventricular end-systolic volume, RV cardiac output, and RVEF. Corresponding values for the left ventricle were also determined. Using long-axis four-chamber, two-chamber, three-chamber, and short-axis slices, the 2D feature-tracking module assessed LV and RV myocardial strain. At the end of diastole in each slice of the series, the endocardial and epicardial outlines were hand-drawn, and the papillary muscles and moderator bands were meticulously removed. The accuracy of feature tracking for endocardial and epicardial contours of the right ventricle was visually evaluated following automated strain analysis on the CMR-FT model. After two observer adjustments, high-quality tracking was achieved for all participants. As stated previously, the Cvi42 calculated and recorded the post-processing for the T2* values [19]. A 1.5T MR scanner and a black-blood gradient echo sequence are used to calculate the T2* of the myocardium. Standard acquisition parameters include a minimum echo time of 2 ms, an echo time interval of 2–3 ms, eight echoes, and a flip angle of 20 degrees. Calculating a monoexponential equation for the signal intensities of the left ventricle (LV) myocardium ROIs at various echo timings yields the T2* value (TEs). On a short-axis image, epicardial and endocardial contours are generated and transmitted to all TEs in order to acquire a myocardial region’s representative signal intensity. T2* values greater than 20 milliseconds (ms) were categorized as normal, 10–20 ms as moderate, and less than 10 ms as severe myocardial iron deposition. Two radiologists independently evaluated the data, and disagreements were resolved by consensus. To examine the agreement between observers, the intraclass correlation coefficient (ICC) with 95% confidence intervals (CI) was calculated. If the ICC value is below 0.5, the reliability is poor. The reliability is moderate if the value falls between 0.5 and 0.75. When the value falls between 0.75 and 0.90, the reliability is good. When the value exceeds 0.90, the reliability is excellent.

## 5. Statistical Analysis

SPSS version 20.0 (SPSS Inc. NORC, Chicago, IL, USA), a statistical program for the social sciences, was used to summarize and analyze the data. Depending on the normality of the data, continuous variables were expressed as the mean standard deviation or medians with interquartile ranges. The Shapiro–Wilk test was used to determine whether the data were normally distributed. The Student’s *t*-test for normally distributed data and the Mann–Whitney U test for non-normally distributed data were used to compare the groups. The correlation between two standard variables was evaluated using Pearson’s correlation coefficient and Spearman’s rank correlation coefficient. The Chi-square test, or Fisher’s exact test, was used to compare categorical variables represented as counts and percentages. Receiver operating characteristics (ROC) analysis and pairwise comparisons of areas under the curves were used to determine the accuracy of patients with MIO classification. The DeLong method compares the regions beneath the curves (AUC). Youden’s test was used to determine the cut-off value. A one-sample t-test was used for normal variables, whereas for non-normal variables, a Wilcoxon signed-rank test was used.

## 6. Results

### 6.1. Cohort Distribution with and without MIO

The retrospective study included 89 individuals with thalassemia major. There were 46 men (51.7%) and 43 women (48.3%), with a mean age of 26.14 ± 7.4 years (range from 10 to 48 years). There were 46 patients without MIO (51.69%) and 43 patients with MIO (48.31%). Thirty-nine patients (32.6%) had severe MIO (T2* Septal ≤ 10 ms), while eighteen patients (19.1%) had mild to moderate MIO (10 ms < T2* Septal < 20 ms) (Figure 1). In total, 45.7% of MIO patients are female, while 54.3% are male. Table 1 lists the baseline characteristics.

### 6.2. Left and Right Ventricular Strain in Relation to Function

The LVEF, LVGRS, LVM, and RVGLS were significantly reduced between the groups with normal and abnormal MIO (T2* Septal < 20 ms) (Table 1). Significant correlations were found between RVEF and T2* values (r = 0.274, *p* = 0.014) and LVEF and T2* values (r = 0.256, *p* = 0.022). LVESV/BSA and LVM were significantly greater in the MIO group compared to the control group (*p* = 0.02). There was no discernible variation in the volumes of the right ventricle (Table 1).

### 6.3. Correlation of T2* with RV/LV Strain Values and Other Blood Parameters

Table 2 displays the correlation between T2* values and age, Hb, platelet count, and RV/LV strain values. T2* values are significantly correlated with the platelet count (r = −0.256, *p* = 0.015), Hb (r = −0.240, *p* = 0.024), and RVGLS (r = −0.257, *p* = 0.02). The correlations were significant despite being weak. In addition, there is no correlation between age and T2* values (r = −0.150, *p* = 0.162). The scatter plots depict the relationship between septal T2* values and RVGLS, LVEF, and RVEF values (Figure 2). The strain levels of the right and left ventricles were not statistically different between the severe (T2* Septal ≤ 10 ms) and mild (T2* Septal < 20 ms) groups.

### 6.4. RVGLS in Thalassemia

With an area under the ROC curve (AUC) of 0.62, an RVGLS with a cut-off value of −20.68% could predict pathological myocardial iron overload (sensitivity = 67.45% and specificity = 58%). Moreover, our findings indicate that an LV EF cut-off value of 56.26% (sensitivity of 79% and specificity of 47.8%) and a Hb cut-off value of 9.35 (sensitivity of 56.5% and specificity of 79%) could predict myocardial iron overload. The *p* values greater than 0.05 calculated using DeLong’s method indicated that the AUC values for LVEF (AUC = 0.65), Hb (AUC = 0.68), and RVGLS (AUC = 0.62) are not statistically significant for identifying MIO. In conjunction with the predictors LVEF, Hb, and RVGLS, a multiple logistic regression model was employed to improve the prediction of abnormal myocardial iron overload. The AUC was 0.783, with a sensitivity of 72% and a specificity of 76% (Figure 3). 

### 6.5. Comparison of Cardiac Function and Strain

Figure 4 depicts a patient with progressive deterioration of cardiac function and the strain analysis that corresponds to it. In individuals with normal MIO, the lower limit for normal RVEF is 53%, whereas the lower limit for normal LVEF is 60%. In this group, 48.3% of the patients had a normal RVEF, and 43.8% had a normal LVEF. Moreover, 45.7% (21/46, *p* = 0.674) of patients with normal RVEF and 32.6% (15/46, *p* = 0.034) of patients with normal LVEF have abnormal MIO (Table 3). The strain values were also compared to the following categories of LV and RV ejection fractions: LVEF > 60% (preserved), LVEF ≤ 60%, RVEF > 53% (preserved), and RVEF ≤ 53%. In addition, 18% of the patients underwent splenectomy, of which 14.3% had abnormal MIO.

### 6.6. Comparison of Myocardial Strain and MIO

The levels of right ventricular and left ventricular strain in the normal MIO and abnormal MIO groups are compared to those of the control group in Table 4. The abnormal MIO groups have significantly lower strain values for LVGCS, RVGRS, RVGCS, and RVGCS- basal compared to the control group. The strain values of LVGRS (*p* = 0.000), RVGRS (*p* = 0.000), RVGCS (*p* = 0.007), and RVGCS-basal (*p* = 0.000) in the category with LVEF > 60% are significantly lower than in the control group, as shown in Table 5. Compared to the normal LV ejection fraction, these abnormal strain parameters indicate the dysfunction of the LV. Moreover, with the exception of RVGLS (*p* = 0.761), all strain values in the category LVEF ≤ 60% are significantly lower than those of the control group with a *p*-value of 0.05. In terms of RVGLS values, there is no statistically significant difference between the control group and the abnormal LVEF group.

### 6.7. Comparison of Myocardial Strain and Right Ventricular Ejection Fraction

Table 6 compares the right ventricle and left ventricle strain values in the normal and abnormal groups based on RVEF to the control group and finds that all strain values in the group with RVEF 53% are significantly different from those in the control group, with the exception of RVGLS (*p* = 0.935) and RVGCS-apex (*p* = 0.076). Additionally, the strain values of LVGRS, RVGRS, and RVGCS-basal are considerably lower than those of the control group (*p* < 0.05) in the category of RVEF > 53% (*p* = 0.001, 0.000, and 0.000, respectively). These abnormal strain parameters suggest that, in patients with preserved RVEF (>53%), strain imaging can more accurately predict the malfunctioning of the RV. Table 7 shows how to use the LV EF, RVGLS, and Hb to predict an abnormal MIO in the clinic.

### 6.8. Inter-Observer Variability

The second observer repeated measurements on ten patients selected at random to assess their repeatability. Table 8 presents the interobserver correlations for the left ventricular and right ventricular strain readings. Table 8 demonstrates that the repeatability between observers for all strain values was excellent.

## 7. Discussion

CMR-FT was utilized in this study to investigate the association between MIO and biventricular function. Our results reveal a substantial relationship between RVEF and T2* levels, as well as between LVEF and T2* values (Table 2). Patients with MIO had significantly lower LVGCS, RVGRS, RVGCS, and RVGCS-basal values, which may be used to predict a T2* less than 20 ms (Table 4). With a sensitivity of 67.4% and a specificity of 58%, RVGLS > −20.68% was one of the most accurate predictors of abnormal MIO. Patients with an RV GLS greater than −20.68% had a significantly increased risk of abnormal MIO (odds ratio, OR = 2.87, 95% CI (1.21, 6.81). Significantly decreased RVGRS and RVGCS-basal strain values were indicative of RV dysfunction in people with preserved RVEF (>53%). (Table 4). Additionally, the LVGRS had a favorable correlation with T2* levels (Table 2). The LVGRS, RVGRS, RVGCS, and RVGCS-basal groups had significantly lower LVEF than the control group, suggesting that individuals with preserved LVEF > 60% are the strongest predictors of left ventricular dysfunction (Table 5). These four strain levels suggested subclinical left ventricular impairment in the presence of preserved LVEF. The RVGRS, RVGCS, and RVGCS-basal groups, with the exception of the LVGRS, were very sensitive and had significantly lower strain parameters and a correlation with T2* than those without MIO (Table 4).

Rezaeian et al. found a statistically significant correlation between LVGLS and LVGCS, but not between LVGRS and T2* [21]. One possible explanation for the divergence in results is that although 70% in the preceding study had MIO, 48.36% had MIO in our study [21]. As comparable observations have previously been documented [7,22], the variance in LVGRS values between groups deserves more explanation. This demonstrates that while MIO is essential, it may not be sufficient to cause global or segmental left ventricular failure. Impairment of LVEF in these patients may occur from the combination of a number of previously described variables [7,22]. In addition to MIO, they have demonstrated that an immune-inflammatory pathway plays a substantial role in causing left ventricular failure in these individuals. Consequently, the strain values (which represent the LV function) and the T2* values (which reflect the MIO function) might not have a linear relationship [7,22]. Two recent studies examining the predictive value of LVGLS in patients with preserved EF found it to be favorable in the majority of cases. In addition to LVGLS, these studies did not include key clinical factors such as NT-proBNP blood levels [23,24]. When Pellicori et al. incorporated blood NT-proBNP levels into their multivariable model, they discovered that LVGLS was no longer associated with hospitalization or death in patients with heart failure [25].

The literature regarding the significance of right ventricular involvement in TM remains scant. Due to the right ventricle’s complicated anatomy and muscle patterns, cardiovascular magnetic resonance (CMR) has been regarded as the gold standard for evaluating the right ventricle’s dependable function and chamber size [6,15,16,17]. We have demonstrated that CMR-derived FT-strain can detect changes in right ventricular myocardial strain induced by subclinical right ventricular dysfunction caused by MIO. RVGLS > −20.68% was one of the most accurate predictors of abnormal MIO, with a sensitivity and specificity of 67.4% and 58.2%, respectively (Figure 2). Rezaeian et al. made a similar observation with RVGLS (−16.4114.73), which had a strong link to T2* [21]. While the RVGLS and MIO demonstrated a substantial link in our cohort, this correlation does not hold true for the diagnosis of subclinical right ventricular dysfunction. The RVGRS, RVGCS, RVGCS-basal, and RVGCS-mid values were considerably lower in the RVEF ≤ 53% group when compared to the RVEF > 53% group (Table 6). However, the values of RVGLS were not significantly lowered in the group with RVEF > 53%. Only RVGRS and RVGCS-basal values were considerably lower in the latter group compared to the other right ventricular strain measures (Table 6). It demonstrates that radial and circumferential values of the right ventricle alter more quickly than longitudinal values in cases of subclinical dysfunction in the RVEF > 53% categories of the TM. This could be because of the complicated ways in which the right ventricle fails in people with HF in TM, or it could be because there aren’t many people in our group with right ventricular dysfunction.

As shown by necropsy [26,27], the right and left ventricles of TM patients were impacted by hypertrophy and myocyte breakdown. Moreover, a right ventricle with a thin wall thickness may be more susceptible to premature failure [28]. Increased volume overload due to chronic anemia, myocardial iron involvement, and cardiomyopathy induced by iron excess on myocytes are suggested pathologies causing right ventricular failure [29]. The accumulation of hemosiderin in the heart affects the action of the mitochondrial respiratory chain, produces ventricular dysfunction, and can lead to heart failure [30]. Because these problems are so common, they may affect how well the right and left ventricles work.

Myocardial iron deposition is also strongly correlated with right ventricular dysfunction, which reflects the decrease in LV function associated with increased myocardial iron loading and lower T2* [31]. The substantial correlations between RVEF and T2* levels and LVEF and T2* values in our cohort support this theory. A retrospective analysis of CMR data from 319 thalassemia significant individuals revealed a gradual decline in right ventricular ejection fraction as myocardial iron burden increased [13]. Iron deposition in the myocardium is also associated with dysfunction of the right ventricle (Figure 1C), which reflects the decline in left ventricular function associated with increased myocardial iron loading and lower T2* [13]. Carpenter et al. also found that people with siderotic cardiomyopathy had up to 21% less iron buildup on the wall of the right ventricle than on the wall of the left ventricle [32]. T2* CMR accurately predicts MIO in thalassemia major, and Anderson et al. [4] discovered a consistent decline in left ventricular ejection fraction (LVEF) with increasing MIO severity (as assessed by T2*). Strain analysis is especially advantageous in HF with preserved RVEF and LVEF because it enables the detection of either a global or localized strain reduction [5,33,34]. In addition, the LVESV/BSA and left ventricular mass of our MIO patients were significantly greater than those with normal MIO.

## 8. Limitations

Our study has a number of limitations. The method was executed in a single-center arrangement with a single-vendor application. The absence of well-established T2* sequences and the inability to consistently analyze them continue to restrict the availability of software to calculate CMR-derived strain parameters. In a preliminary investigation, we were able to demonstrate that RVGLS may have predictive utility. However, this early experiment was conducted on a modest scale, with only 89 participants. Our study is limited by the small sample size of patients with MIO; hence, additional prospective studies with larger study populations are required. A follow-up study is necessary to determine how these individuals’ symptoms progressed from asymptomatic to clinical HF/LV failure. Additionally, biochemical markers such as B-type natriuretic peptides, which are thought to be the best way to find subclinical cardiac dysfunction, were not looked into.

## 9. Conclusions

RVGLS was the most accurate predictor of myocardial iron overload. CMR-FT was able to detect significant decreases in RVGLS, RVGRS, and RVGCS-basal strains in TM patients with preserved RVEF, allowing us to characterize these patients as having right ventricular dysfunction despite their preserved ejection fraction. In the majority of TM patients, a decreased RVEF was associated with MIO. FT-strain will contribute to our pathophysiology understanding of how MIO affects the myocardium, although T2* will remain the diagnostic gold standard. In addition, serial strain analysis can assist in finding therapeutic strategies to increase the time between the onset of subclinical strain change in the right or left ventricle and the onset of symptomatic heart failure in asymptomatic individuals.

## Figures and Tables

**Figure 1 diagnostics-12-01920-f001:**
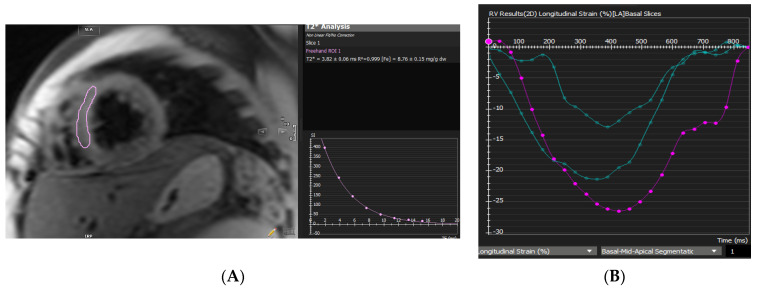
A 16-year-old boy with thalassemia major. (**A**) MIO (T2* 3.8 ms); (**B**) RV GLS; (**C**) GRS.

**Figure 2 diagnostics-12-01920-f002:**
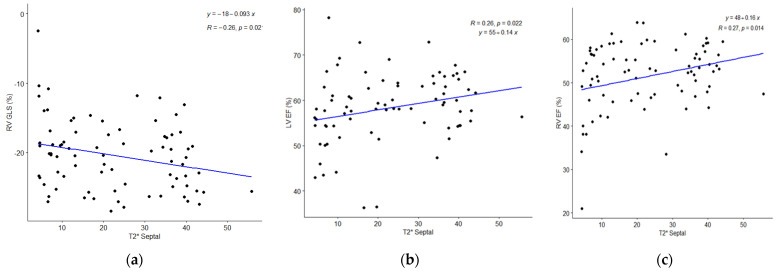
Depicts the relationship between T2*Septal measurements and the RV GLS, LV, and RV Ejection Fractions using a scatter plot. (**a**) RVGLS; (**b**) LVEF; (**c**) RVEF.

**Figure 3 diagnostics-12-01920-f003:**
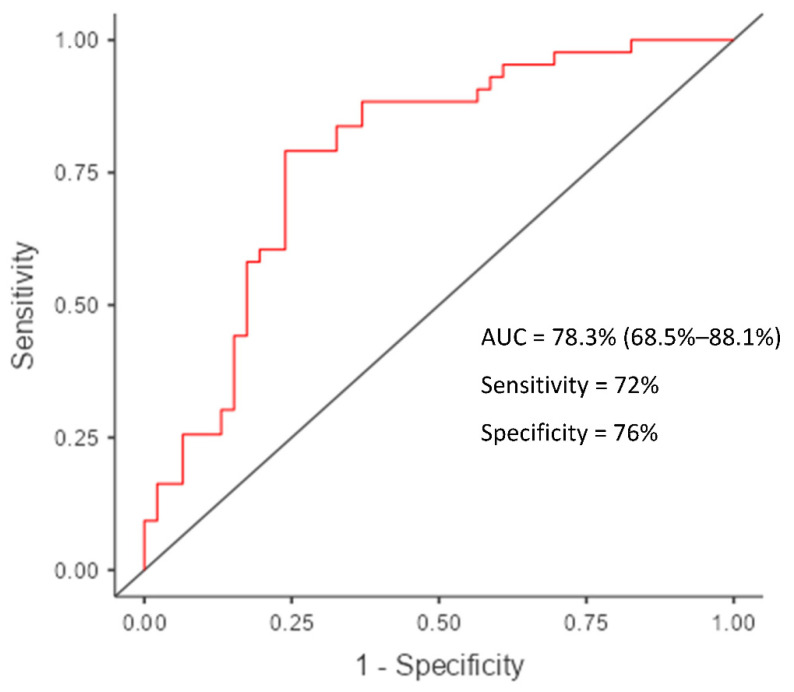
ROC curve for predicting myocardial iron overload (T2*Septal level 20) using a multiple logistic regression model with predictors Hb, LVEF, and RVGLS.

**Figure 4 diagnostics-12-01920-f004:**
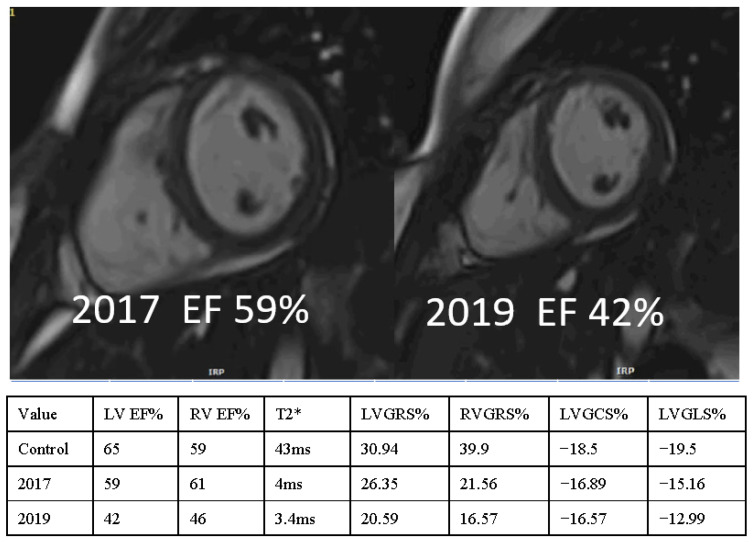
A 19-yearold-girl with thalassemia major with deterioration of the cardiac function due to increased MIO. The details of the parameters are noted in the attached table with a comparison of their respective values.

**Table 1 diagnostics-12-01920-t001:** Comparison of baseline characteristics in normal and abnormal MIO groups.

Variable	Normal MIO *(n* = 43)	Abnormal MIO (*n* = 46)	Overall (*n* = 89)	*p*-Value	
Age	26 ± 8.59	28 ± 7.63	27 ± 8.12	0.29	
Hb	9.10 (0.50)	9.40 (0.975)	9.20 (0.70)	0.00 *	
Platelet count (10^3^/μL)	266 (19.5)	278 (86.0)	269 (28.0)	0.03 *	
LVEF (%)	60.38 ± 5.32	55.99 ± 9.11	58.24 ± 7.68	0.01 *	
LVEDV (mL)	126.9 ± 34.69	129.4 ± 34.67	128.1 ± 34.48	0.57	
LVESV (mL)	46.3 (22.99)	52.2 (24.86)	49.7 (24.70)	0.05	
LVEDV/BSA (mL/m^2^)	80.3 ± 14.51	84.0 ± 19.31	82.1 ± 17.01	0.13	
LVESV/BSA (mL/m^2^)	29.7 (8.15)	36.2 (10.45)	31.8 (10.70)	0.02 *	
RVEDV (mL)	114.3 (87.23)	128.7 (52.21)	125.2 (61.10)	0.79	
RVESV (mL)	53.1 (40.38)	60.3 (28.57)	58.2 (33.80)	0.53	
RVEDV/BSA (mL/m^2^)	76.9 (34.24)	83.8 (19.15)	82.5 (26.30)	0.96	
RVESV/BSA (mL/m^2^)	38 (14.98)	40.3 (11.55)	38.6 (15.30)	0.42	
LVGCS (%)	−18.5 (3.05)	−17.4 (4.12)	−18.2 (3.44)	0.37	
LVGLS (%)	−17.9 (3.98)	−15.8 (2.97)	−16.8 (3.80)	0.18	
LVGRS (%)	30.94 ± 5.38	28.21 ± 5.99	29.61 ± 5.82	0.03 *	
LVM (gm)	44.39 ± 7.79	49.25 ± 11.00	46.76 ± 9.74	0.02 *	
RVEF (%)	53.3 (9.45)	52.5 (10.8)	52.8 (9.86)	0.65	
RVGCS (%)	−12.29 ± 1.76	−12.16 ± 3.26	−12.23 ± 2.59	0.82	
RVGCS-apex (%)	−13.91 ± 3.31	−14.26 ± 4.03	−14.08 ± 3.66	0.67	
RVGCS-basal (%)	−10.79 ± 2.67	−10.55 ± 2.66	−10.67 ± 2.65	0.68	
RVGCS-mid (%)	−13.44 ± 2.22	−13.33 ± 3.67	−13.39 ± 2.99	0.87	
RVGLS (%)	−21.7 (7.72)	−19.4 (6.95)	−20.3 (7.23)	0.04 *	
RVGRS (%)	19.89 ± 3.65	19.52 ± 6.24	19.70 ± 5.05	0.74	

Values are expressed as mean ± SD or median (Interquartile Range) depending upon normality of data. Abbreviations: LVEF—left ventricular ejection fraction; RVEF—right ventricular ejection fraction; GRS—global radial strain; GCS—global circumferential strain; GLS—global longitudinal strain; * *p* < 0.05, statistically significant difference.

**Table 2 diagnostics-12-01920-t002:** Correlation between T2* Septal measurements and Age, Hb, platelet count, and RV/LV strain values.

Variables	T2 * Septal	*p*-Value
	Correlation Coefficient	
Age	−0.150	0.162
Hb	−0.240	0.024 *
Platelet count (10^3^/μL)	−0.256	0.015 *
LVEF (%)	0.256	0.022 *
RVEF (%)	0.274	0.014 *
LVGRS (%)	0.232	0.038 *
LVGCS (%)	−0.217	0.053
LVGLS (%)	−0.194	0.085
RVGRS (%)	0.046	0.685
RVGCS (%)	−0.029	0.797
RVGLS (%)	−0.257	0.021 *
RVGCS-basal (%)	−0.054	0.632
RVGCS-mid (%)	0.018	0.877
RVGCS-apex (%)	0.070	0.54

* *p* < 0.05, statistically significant difference.

**Table 3 diagnostics-12-01920-t003:** Percentages of patients with normal and abnormal MIO in categories based on sex, splenectomy, deceased, normal LV and RV ejection fractions.

Variables		MIO		*p*-Value
		Abnormal	Normal	
Sex	Female	21 (48.8%)	22 (51.2%)	0.67
	Male	25 (54.3%)	21 (45.7%)	
Had Splenectomy	No	33 (45.2%)	40 (54.8%)	0.01 *
	Yes	13 (81.2%)	3 (18.8%)	
Deceased	No	45 (52.3%)	41 (47.7%)	0.61
	Yes	1 (33.3%)	2 (66.7%)	
LVEF	Abnormal (≤60%)	31 (67.4%)	19 (44.2%)	0.03 *
	Normal (>60%)	15 (32.6%)	24 (55.8%)	
RVEF	Abnormal (≤53%)	25 (54.3%)	21 (48.8%)	0.67
	Normal (>53%)	21 (45.7%)	22(51.2%)	

** p* < 0.05, statistically significant difference.

**Table 4 diagnostics-12-01920-t004:** Comparison of RV and LV strain values in normal and abnormal MIO groups with the control group.

Variable	Control Group (*n* = 24) Mean ± SD/ Median (IQR)	Abnormal MIO (*n* = 46) Mean ± SD/ Median (IQR)	*p*-Value	Normal MIO (*n* = 43) Mean ± SD/ Median (IQR)	*p*-Value
LVGRS (%)	29.70 (6.06)	28.60 (9.30)	0.10	30.93 (7.99)	0.94
LVGCS (%)	−19.00 ± 1.79	−17.43 ± 2.44	0.00 *	−18.41 ± 2.13	0.41
LVGLS (%)	−17.90 (1.50)	−16.30 (2.88)	0.08	−17.90 (3.98)	0.93
RVGRS (%)	39.90 (12.00)	21.50 (10.90)	0.00 *	20.00 (4.75)	0.00 *
RVGCS (%)	−14.10 ± 2.25	−12.12 ± 3.19	0.00 *	−12.36 ± 1.85	0.00 *
RVGLS (%)	−20.30 (2.15)	−20.14 (6.09)	0.36	−21.71 (7.72)	0.35
RVGCS-basal (%)	−13.60 ± 2.00	−10.70 ± 2.68	0.00 *	−10.84 ± 2.65	0.00 *
RVGCS-mid (%)	−14.10 ± 2.41	−13.20 ± 3.64	0.23	−13.52 ± 2.27	0.11
RVGCS-apex (%)	−15.30 (7.87)	−14.90 (5.00)	0.33	−14.43 (4.89)	0.11

* *p* < 0.05, statistically significant difference.

**Table 5 diagnostics-12-01920-t005:** Comparison of RV and LV strain values in normal and abnormal groups based on LVEF with the control group.

Train Variables	Control Group (*n* = 24)	LVEF ≤ 60% (*n* = 50)	*p*-Value	LVEF > 60% (*n* = 39)	*p*-Value
	Mean ± SD/	Mean ± SD/		Mean ± SD/	
	Median (IQR)	Median (IQR)		Median (IQR)	
LVGRS (%)	29.70 (6.06)	27.08 (6.93)	0.00 *	34.42 (6.93)	0.00 *
LVGCS (%)	−19.00 ± 1.79	−16.90 ± 2.14	0.00 *	−19.23 ± 1.87	0.07
LVGLS (%)	−17.90 (1.50)	−15.91 (3.12)	0.00 *	−18.53 (3.44)	0.78
RVGRS (%)	39.90 (12.00)	20.06 (8.44)	0.00 *	20.50 (8.22)	0.00 *
RVGCS (%)	−14.10 ± 2.25	−11.72 ± 2.62	0.00 *	−12.91 ± 2.90	0.00 *
RVGLS (%)	−20.30 (2.15)	−20.41 (5.48)	0.76	−19.54 (8.01)	0.57
RVGCS-basal (%)	−13.60 ± 2.00	−10.19 ± 2.81	0.00 *	−11.44 ± 2.29	0.00 *
RVGCS-mid (%)	−14.10 ± 2.41	−12.67 ± 2.89	0.00 *	−14.27 ± 3.03	0.76
RVGCS-apex (%)	−15.30 (7.87)	−14.78 (3.97)	0.04 *	−13.92 (6.26)	0.41

****p* < 0.05, statistically significant difference.

**Table 6 diagnostics-12-01920-t006:** Comparison of RV and LV strain values in normal and abnormal groups based on RV EF with the control group.

Strain Variables	Control Group (*n =* 24)	RVEF ≤ 53% (*n* = 46)	*p*-Value	RVEF > 53% (*n* = 43)	*p*-Value
	Mean ± SD/	Mean ± SD/		Mean ± SD/	
	Median (IQR)	Median (IQR)		Median IQR)	
LVGRS (%)	29.70 (6.06)	27.26 (7.55)	0.00 *	32.69 (7.23)	0.00 *
LVGCS (%)	−19.00 ± 1.79	−17.05 ± 2.51	0.00 *	−18.82 ± 1.73	0.46
LVGLS (%)	−17.90 (1.50)	−15.96 (2.71)	0.00 *	−18.10 (3.69)	0.94
RVGRS (%)	39.90 (12.0)	17.98 (6.73)	0.00 *	22.54 (5.35)	0.00 *
RVGCS (%)	−14.10 ± 2.25	−10.93 ± 2.36	0.00 *	−13.63 ± 2.13	0.19
RVGLS (%)	−20.30 (2.15)	−20.18 (4.77)	0.94	−20.70 (8.15)	0.56
RVGCS-basal (%)	−13.60 ± 2.00	−9.74 ± 2.62	0.00 *	−11.80 ± 2.28	0.00 *
RVGCS-mid (%)	−14.10 ± 2.41	−12.08 ± 2.69	0.00 *	−14.76 ± 2.79	0.14
RVGCS-apex (%)	−15.30 (7.87)	−13.66 (5.30)	0.07	−15.28 (4.29)	0.51

* *p* < 0.05, statistically significant difference.

**Table 7 diagnostics-12-01920-t007:** Clinical Utility of LVEF, RVGLS, and Hb as predictors in assessing MIO.

		95% Confidence Interval	
Decision Statistics	Estimate	Lower	Upper
Apparent prevalence	48.3%	37.6%	59.2%
True prevalence	48.3%	37.6%	59.2%
Test sensitivity	74.4%	58.8%	86.5%
Test specificity	76.1%	61.2%	87.4%
Diagnostic accuracy	75.3%	65.0%	83.8%
Positive predictive value	74.4%	58.8%	86.5%
Negative predictive value	76.1%	61.2%	87.4%
Proportion of false positives	23.9%	12.6%	38.8%
Proportion of false negative	25.6%	13.5%	41.2%

**Table 8 diagnostics-12-01920-t008:** Reproducibility of data using intraclass correlation coefficients.

Variables	Inter-Observer Reproducibility
LVGLS	0.996 (0.968–0.999)
LVGRS	0.991 (0.955–0.998)
LVGCS	0.986 (0.948–0.996)
RVGLS	0.994 (0.978–0.998)
RVGRS	0.997 (0.990–0.999)
RVGCS	0.967 (0.856–0.993)

Data are represented using intraclass correlation coefficients with 95% CI.

## Data Availability

Available with the corresponding author.

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
