# Peer review of "Significance of Cardiac Magnetic Resonance Feature Tracking of the Right Ventricle in Predicting Subclinical Dysfunction in Patients with Thalassemia Major"

_diagnostics, 2022, doi:10.3390/diagnostics12081920_

Round 1
Reviewer 1 Report
The aim of this study was to determine the role of CMR-FT of the right ventricle for the prediction of sublicinal dysfunction in TM patients.
The study is really interesting, but in its current form is really difficult to read and I strongly recommend an an English revision .
Introduction:
- Only thalassemia major patients needs blood transfusions. Thalassemia intermedia patients are often not regularly transfused. Please, rephrase.
- Not only children, but also adult patients with TM need transfusions.
- The last paragraph of the Discussion, describing the CMR-FT, should be transferred in the Introduction.
- The study corresponding to reference 8 is not recent (2010).
Methods:
- In high-output state, ejection fraction is typically normal or high. Please, correct.
- You should better clarify how the ranges for LV and RV EF were obtained. Mean-2SD? Percentiles?
- RVGS was measured only in the septum or is representative of the whole myocardium?
- How did you obtain RV GSG apex, basal, and mid?
- You should briefly describe how T2* value was measured.
- Since many of your variables were not normally distributed, you should use Spearman’s correlation.
- You described the consensus between operators in both “cardiac MR acquisition” and “statistical Analysis” sections. Please, avoid the repetitions.
Results.
- Divide the Results section in different sub-sections.
- Use always the same number of decimal places (1 or 2).
- Generally, Hb values are not associated with cardiac T2* values. How can you explain this finding?
- It is really surprising that almost half of your patients had a pathological cardiac T2*. Were all patients chelated? Briefly describe the chelation therapy and the compliance.
- You should clearly state that, although the correlation between cardiac T2* and LV and RV EF and RVGLS was significant, the correlation coefficient was really weak.
- Results of ROC analysis (sensitivity and specificity) should be reported in the Results section, and not in the Discussion.
- Figure 4: what does Control mean? It is the first CMR scan?
- While in Table 1 the LV GRS (as well as other variables) is described as mean ± SD and in table 4 as median (IQR)? This is really confounding. Use always the same statistical description for each variable.
- There are several discrepancies in the reported data. For example, the RVGS in pts with abnormal MIO is -12.16±3.26 in Table 1 and -12.10±3.19 in Table 4. You should check all data.
- Specify the thresholds detected for Hb and LV EF in predicting MIO.
- You calculated also hepatic T2* values, but they had not been considered in the analysis.
Discussion.
- The Discussion needs to be better organized.
- There are different repetitions, that should be eliminated.
Author Response
Dear Honorable reviewer: We would like to thank you for the suggestion. We have revised the whole manuscript as per the suggestion and incorporated the changes meticulously. Should there be any shortcomings we would be delighted to address them again.
Comments and reply
The study is really interesting, but in its current form is really difficult to read and I strongly recommend an an English revision . Reply: English revision done
Introduction:
- Only thalassemia major patients needs blood transfusions. Thalassemia intermedia patients are often not regularly transfused. Please, rephrase.: Reply: The sentence is rephrased.
- Not only children, but also adult patients with TM need transfusions. Reply: The sentence is rephrased.
- The last paragraph of the Discussion, describing the CMR-FT, should be transferred in the Introduction.:
Reply:The last paragraph has been incorporated into the the introduction section.
- The study corresponding to reference 8 is not recent (2010).:
Reply: An additional recent reference has been added. The present reference is 13 and 14.
Methods:
- In high-output state, ejection fraction is typically normal or high. Please, correct.:
Reply: The sentence is rephrased.
- You should better clarify how the ranges for LV and RV EF were obtained. Mean-2SD? Percentiles?
Reply: The 95% Confidence Interval for the mean LV EF was (56.86, 60.04) and the 95% Confidence Interval for the mean RV EF was (49.77, 53.24). Based on these upper bounds of the means of LV and RVEF, we categorized the patients into groups, normal and abnormal as discussed in the manuscript.
- RVGS was measured only in the septum or is representative of the whole myocardium? : Reply: whole myocardium
- How did you obtain RV GSG apex, basal, and mid?: Reply:All three apex, basal, and mid GCS was measured by the same CVI 42 software.
- You should briefly describe how T2* value was measured.: Reply: Described in the method section.
- Since many of your variables were not normally distributed, you should use Spearman’s correlation.
Reply: Incorporated the suggestion in line 117.
- You described the consensus between operators in both “cardiac MR acquisition” and “statistical Analysis” sections. Please, avoid repetitions. Reply: Corrected
Results.
- Divide the Results section into different sub-sections.: Reply: Divisions are added.
- Use always the same number of decimal places (1 or 2).: Reply: Corrected.
- Generally, Hb values are not associated with cardiac T2* values. How can you explain this finding?
Reply:A RVGLS with a cut off value -20.68% could predict pathological myocardial iron overload with the area under the ROC curve (AUC) = 0.62 (Sensitivity = 67.45% and Specificity = 58%). Also, our analysis suggests that a LV EF with a cut-off value 56.26 % (sensitivity of 79% and specificity of 47.8%) and Hb with a cut-off value 9.35 (sensitivity of 56.5% and specificity of 79%) could predict myocardial iron overload. The P values > 0.05 calculated using DeLong's approach indicated that the AUC values for LVEF (AUC = 0.65), Hb (AUC = 0.68) and RVGLS (AUC = 0.62) are not statistically significant differ in identifying MIO. A multiple logistic regression model was employed in conjunction with the predictors LVEF, Hb, and RVGLS to enhance the prediction of abnormal myocardial iron overload. The AUC was 0.783 with 72% sensitivity and 76% specificity (Figure 3).
- It is really surprising that almost half of your patients had a pathological cardiac T2*. Were all patients chelated? Briefly describe the chelation therapy and the compliance.
Reply: Patients are treated in Dubai Thalassemia Center, all chelators are available including Deferasirox, Desferoxamine, and Deferasirox the main underlying reason for uncontrolled iron overload is patients' compliance to chelation therapy.
- You should clearly state that, although the correlation between cardiac T2* and LV and RV EF and RVGLS was significant, the correlation coefficient was really weak.
Reply: The correlation between T2* values and Age, Hb, Platelet Count, and RV / LV strain values are reported in Table 2. The platelet count (r = -0.256, p = 0.015), Hb (r = -0.240, p = 0.024) and RVGLS (r = -0.257, p = 0.02) values correlate significantly with T2* values. Although the correlations were weak, they were significant.
- Results of the ROC analysis (sensitivity and specificity) should be reported in the Results section, and not in the Discussion. Reply: Removed from the discussion section.
- Figure 4: what does Control mean? It is the first CMR scan?: Reply: It is explained in the method section.
- While in Table 1 the LV GRS (as well as other variables) is described as mean ± SD and in table 4 as median (IQR). This is really confounding. Use always the same statistical description for each variable.
Reply: In Table 1 the values are reported as mean ± SD and in table 4 as median (IQR). This is based on the normality of the data in each group we are comparing. Accordingly, the statistical methods we are using for the comparison will change. For example, if the normality assumption holds, we conduct t-test, otherwise, we perform the nonparametric test.
- There are several discrepancies in the reported data. For example, the RVGS in pts with abnormal MIO is -12.16±3.26 in Table 1 and -12.10±3.19 in Table 4. You should check all data. Reply: Corrected.
- Specify the thresholds detected for Hb and LV EF in predicting MIO.
Reply: We have updated the thresholds for Hb and LV EF. “ Also, our analysis suggests that an LV EF with a cut-off value of 56.26 % (sensitivity of 79% and specificity of 47.8%) and Hb with a cut-off value of 9.35 (sensitivity of 56.5% and specificity of 79%) could predict myocardial iron overload.”
- You calculated also hepatic T2* values, but they had not been considered in the analysis.: Reply:We have deleted the same as we have not included in the analysis.
Discussion.
- The Discussion needs to be better organized.: Reply: Organized.
- There are different repetitions, that should be eliminated. : Reply: Corrected.

Reviewer 2 Report
In the manuscript entitled "Significance of Cardiac magnetic resonance feature tracking of the right ventricle in predicting subclinical dysfunction in patients with thalassemia major" Karuna M Das et al. provide information regarding the usefulness of cardiac magnetic resonance feature tracking for prediction purposes in the background of thalassemia major. The manuscript is well structured, the study uses an average cohort of patients and the authors discuss the limitations of their study. However, this reviewer has some comments, as follow:
1. A general comment: this reviewer understands that most of the abbreviations used throughout the manuscript are common but some phrases in the results section are too hard to follow due to their excessive use. I believe that keeping the use of abbreviation as low as possible will benefit the results section and will make it easier to follow.
2. Results section, lines 148-151: the authors give two different AUC values for RVGLS. Did the authors apply different statistical methods to assess AUC levels? Can you please explain?
3. Discussion section, lines 195-197: the authors comment on the prediction potential of RVGLS. However, the AUC (0.618) is rather low and this reviewer believes that a comment regarding the power of this predictor should be added to complete the sentence.
4. Please proof-read the manuscript for some grammatical mistakes in the main body and the figure legends.
Author Response
Dear Honorable reviewer: We would like to thank you for the suggestion. We have revised the whole manuscript as per the suggestion and incorporated the changes meticulously. Should there be any shortcomings we would be delighted to address them again.
Comments and reply
. A general comment: this reviewer understands that most of the abbreviations used throughout the manuscript are common but some phrases in the results section are too hard to follow due to their excessive use. I believe that keeping the use of abbreviation as low as possible will benefit the results section and will make it easier to follow. : Reply: The abbreviations are minimized.
- Results section, lines 148-151: the authors give two different AUC values for RVGLS. Did the authors apply different statistical methods to assess AUC levels? Can you please explain?
Reply: AUC value for RVGLS is 0.618. The two different AUC values reported in the manuscript was a typo. We have corrected the same.
- Discussion section, lines 195-197: the authors comment on the prediction potential of RVGLS. However, the AUC (0.618) is rather low and this reviewer believes that a comment regarding the power of this predictor should be added to complete the sentence.
Reply: The patients with RV GLS > −20.68% had a significantly increased risk of showing abnormal MIO (odds ratio, OR = 2.87 with 95% CI of (1.21, 6.81))
- Please proof-read the manuscript for some grammatical mistakes in the main body and the figure legends.
Reply: We have incorporated the suggestions

Reviewer 3 Report
Design: observational retrospective study in 89 patients
The purpose of this paper is to evaluate if CMR strain could be used to detect RV dysfunction and to measure RV myocardial deformation in patients with TM who had a preserved RVEF.
Comments as follows:
1. Any correlations with late gadolinium enhancement of LV or RV?
2. Did you performed also T1 mapping in these patients?
Author Response
Dear Honorable reviewer: We would like to thank you for the suggestion. We have revised the whole manuscript as per the suggestion and incorporated the changes meticulously. Should there be any shortcomings we would be delighted to address them again.
Comments and reply
- Any correlations with late gadolinium enhancement of LV or RV? Reply: : unfortunately it was not done.
- Did you performed also T1 mapping in these patients? Reply: unfortunately, it was not done

Round 2
Reviewer 1 Report
The authors have addressed all my issues.
This manuscript is a resubmission of an earlier submission. The following is a list of the peer review reports and author responses from that submission.